# Biological age threshold is associated with symptomatic knee osteoarthritis risk in chinese adults: Insights from machine learning analysis of a national cohort

Fanyu Fu[1�], Li Dong[2�], Jiwei Lian[3], Chang Liu[4], Tingting Pang[1], Yunli Wang[1], Peng Liu[3]*, Yufeng Wang ⓘ[5]*

1 College of Acupuncture and Tuina, Changchun University of Chinese Medicine, Changchun, Jilin Province, China, 2 College of Rehabilitation Medicine, Changchun University of Chinese Medicine, Changchun, Jilin Province, China, 3 Department of Tuina, Shenzhen Hospital (Futian) of Guangzhou University of Chinese Medicine, Shenzhen, Guangdong, China, 4 Department of Tuina, Guang'anmen Hospital, China Academy of Chinese Medical Sciences, Beijing, China, 5 Changchun University of Chinese Medicine, Changchun, Jilin Province, China

উ These authors contributed equally to this work.
* 515900837@qq.com (PL); wangchn@126.com (YW)

## Abstract

### Background

Symptomatic knee osteoarthritis (KOA) imposes a substantial global health and economic burden. Although chronological age (CA) is a key risk factor, it poorly reflects interindividual aging heterogeneity. Biological age (BA), which is quantified using blood biomarkers that reflect systemic physiological integrity, is a superior measure of functional decline and molecular aging. Mechanistically, BA may be linked to KOA pathogenesis via cellular senescence and senescence-associated secretory phenotype (SASP).

### Objective

This study aimed to explore the association between BA and symptomatic KOA in a nationally representative Chinese cohort and to evaluate BA's utility of BA in enhancing KOA risk assessment.

### Methods

We conducted a cross-sectional analysis using the 2011/2015 China Health and Retirement Longitudinal Study (CHARLS) data of 1,000 participants (≥45 years old) with complete BA and symptomatic KOA data (defined as self-reported physician-diagnosed osteoarthritis with concurrent knee pain). BA was calculated using the Klemera-Doubal method (KDM) and eight serum biomarkers. Associations were assessed using multivariable-adjusted logistic regression, restricted cubic splines

**Data availability statement:** The data used in this study are provided as Supporting Information file 'Raw Data' and hosted at the following address: https://data.mendeley.com/datasets/3rv7mf5pv9/1.

**Funding:** This study was supported by grants from the Sanming ProjectMedicine in Shenzhen (No.SZZYSM202402015) and Science and Technology Development Project of Jilin Province, China (No.20240304086SF).

**Competing interests:** The authors have declared that no competing interests exist.

(RCS), and subgroup analyses. Six machine learning models (including XGBoost and LightGBM) were used to distinguish cases of symptomatic KOA, with SHAP interpreting the optimal model.

## Results

Participants with symptomatic KOA had a significantly higher mean BA than those without (59.97 vs. 58.76 years, $p < 0.001$). After multivariable adjustment, each 1-year BA increase was associated with 1.23% higher symptomatic KOA odds ($OR$=1.0123, 95% $CI$:1.0049–1.0197, $p$=0.0010). Compared with the lowest BA quartile (Q1), the highest quartiles (Q3 and Q4) showed a significantly elevated symptomatic KOA risk (Q3: $OR$=1.4655, 95% $CI$:1.1989–1.7940, $p$=0.0002; Q4: $OR$=1.4519, 95% $CI$:1.1755–1.7956, $p$=0.0001). RCS analysis revealed a non-linear relationship, with symptomatic KOA risk accelerating beyond approximately 66.7 years ($p$ for non-linearity=0.013). Subgroup analyses demonstrated consistent results. The XGBoost model demonstrated the highest discriminative performance (AUROC=0.9078), with SHAP identifying BA as the most influential feature.

## Conclusion

BA is strongly and non-linearly associated with symptomatic KOA risk in Chinese adults, accelerating beyond a critical threshold. BA assessment may enhance KOA risk stratification and could inform future interventional studies. However, the cross-sectional design of this study precludes causal inferences. Longitudinal studies are required to establish temporal relationships and explore potential causal associations.

## Introduction

Knee osteoarthritis (KOA), a prevalent and debilitating chronic joint disorder characterized by articular cartilage degradation, synovitis, and subchondral bone remodeling, imposes a substantial burden on global healthcare [1]. Established risk factors, including chronological aging, obesity, joint trauma, inflammation, and genetic predisposition, significantly contribute to its pathogenesis and progression [2]. Globally, KOA ranks as the 11th leading cause of disability, affecting approximately 3.8% of the world's population [3]. In China, an estimated 37.35 million adults aged ≥60 years are affected by symptomatic KOA [4], highlighting the urgent need for improved risk stratification strategies to enable early identification and timely intervention.

Although chronological age (CA) is a well-established risk factor for KOA, it inadequately captures the considerable heterogeneity observed in individual aging trajectories and susceptibility to age-related pathologies [5]. Biological age (BA), constructed using panels of accessible blood biomarkers that reflect systemic physiological integrity, including inflammatory, metabolic, and hormonal profiles, provides a more precise quantification of an individual's functional decline and molecular aging processes

than CA alone [6,7]. Globally, various methods have been developed to estimate BA, such as the Klemera-Doubal method (KDM), PhenoAge, and others, each incorporating distinct biomarker panels to capture multidimensional aspects of aging [8,9]. Studies across different populations have shown that accelerated BA, constructed using diverse methodologies, is consistently associated with a higher risk of age-related diseases, including osteoarthritis [10,11]. Mechanistically, accelerated BA is linked to KOA pathogenesis through processes such as cellular senescence and the proinflammatory senescence-associated secretory phenotype (SASP) [12]. Therefore, integrating BA into KOA risk assessment frameworks may improve the understanding of heterogeneous aging-related risk.

The China Health and Retirement Longitudinal Study (CHARLS) provides a valuable resource for constructing predictive models [13]. This nationally representative survey encompassed individuals aged ≥ 45 years in China and gathered extensive information on health, social, and economic variables [14]. Despite the theoretical promise of integrating BA into symptomatic KOA risk assessment, evidence from large-scale, nationally representative studies on the Chinese population remains limited. Moreover, comparative insights from international cohorts regarding BA construction methods and their clinical applicability in the context of KOA are scarce. To address this gap, we conducted a cross-sectional study using data from the 2011 and 2015 waves of CHARLS. We employed descriptive statistics, multivariable logistic regression, restricted cubic spline analysis, and subgroup analyses to examine the association between BA and symptomatic KOA. Furthermore, we developed six machine learning models and used SHapley Additive exPlanations (SHAP) to interpret the optimal model. This study aimed to provide robust evidence for incorporating BA into symptomatic KOA risk assessment, facilitating the identification of high-risk individuals and informing effective prevention strategies.

## Methods

### Study design and data source

This cross-sectional study used data from the CHARLS, a nationally representative cohort of Chinese adults aged ≥ 45 years. We combined data from the 2011 and 2015 survey waves, yielding 38,800 participants in total. All data were downloaded from the official website (https://charls.pku.edu.cn/index.htm) upon requests. CHARLS was approved by the Ethical Review Committee of Peking University (IRB 00001052–11015) [15]. All participants in CHARLS provided written informed consent during data collection. As this study is a secondary analysis of existing publicly available data, no separate ethical approval was required, in accordance with the guidelines of CHARLS. All methods were performed in accordance with the relevant guidelines and regulations of the official CHARLS website [16].

### Study population

Data from the 2011 and 2015 waves of CHARLS were accessed on June 1, 2025, for research purposes. The authors did not have access to information that could identify individual participants, as the CHARLS data used in this study were de-identified and publicly available. The analytical sample was progressively refined from the merged 2011/2015 CHARLS dataset (n = 38,800). We applied the following exclusion criteria: 1) participants aged <45 years (n = 1,787), 2) missing data required to calculate BA (n = 15,394), and 3) missing symptomatic KOA status information (n = 12,114). After these exclusions, 9,505 participants with complete exposure (BA) and outcome (symptomatic KOA) data remained and constituted the final analytical cohort for baseline characterization and primary analysis (Table 1).

From this cohort, 1,000 individuals meeting the diagnostic criteria for symptomatic KOA (self-reported physician-diagnosed osteoarthritis with concurrent knee pain) comprised the case group for subsequent association and predictive modeling. No additional restrictions were applied to this subgroup. Although the exclusion of participants with incomplete data reduced the absolute sample size, the remaining 9,505 participants still reflected the broad demographic and clinical distributions of middle-aged and older Chinese adults in the CHARLS cohort, helping to ensure the external validity of our findings for aging-related epidemiologic research (Fig 1). For the remaining covariates included in the regression

**Table 1. Baseline characteristics of the eligible CHARLS participants with BA and symptomatic KOA status (n=9,505).**

| Variable | Overall | KOA No | Yes | p-value |
|---|---|---|---|---|
| | N=9,505 | N=8,505 | N=1,000 | |
| Age, mean (sd) | 59.53 (9.44) | 59.37 (9.48) | 60.87 (9.04) | <0.001 |
| BMI, mean (sd) | 23.53 (3.84) | 23.50 (3.80) | 23.71 (4.12) | 0.122 |
| BA, mean (sd) | 58.89 (9.96) | 58.76 (9.98) | 59.97 (9.70) | <0.001 |
| Gender, n (p%) | | | | <0.001 |
| Male | 4,414.00 (46.44%) | 4,079.00 (47.96%) | 335.00 (33.50%) | |
| Female | 5,091.00 (53.56%) | 4,426.00 (52.04%) | 665.00 (66.50%) | |
| Marital, n (p%) | | | | 0.179 |
| Living with a partner | 7,936.00 (83.49%) | 7,116.00 (83.67%) | 820.00 (82.00%) | |
| Living alone | 1,569.00 (16.51%) | 1,389.00 (16.33%) | 180.00 (18.00%) | |
| Education, n (p%) | | | | <0.001 |
| Illiterate | 2,805.00 (29.51%) | 2,428.00 (28.55%) | 377.00 (37.70%) | |
| Junior high school or below | 5,779.00 (60.80%) | 5,197.00 (61.11%) | 582.00 (58.20%) | |
| High school or above | 921.00 (9.69%) | 880.00 (10.35%) | 41.00 (4.10%) | |
| Residence, n (p%) | | | | <0.001 |
| Rural | 6,175.00 (64.97%) | 5,411.00 (63.62%) | 764.00 (76.40%) | |
| Urban | 3,330.00 (35.03%) | 3,094.00 (36.38%) | 236.00 (23.60%) | |
| Hypertension, n (p%) | | | | <0.001 |
| No | 7,034.00 (74.00%) | 6,339.00 (74.53%) | 695.00 (69.50%) | |
| Yes | 2,471.00 (26.00%) | 2,166.00 (25.47%) | 305.00 (30.50%) | |
| Dyslipidemia, n (p%) | | | | 0.029 |
| No | 8,612.00 (90.60%) | 7,725.00 (90.83%) | 887.00 (88.70%) | |
| Yes | 893.00 (9.40%) | 780.00 (9.17%) | 113.00 (11.30%) | |
| Diabetes, n (p%) | | | | 0.220 |
| No | 8,941.00 (94.07%) | 8,009.00 (94.17%) | 932.00 (93.20%) | |
| Yes | 564.00 (5.93%) | 496.00 (5.83%) | 68.00 (6.80%) | |
| Cancer, n (p%) | | | | 0.114 |
| No | 9,423.00 (99.14%) | 8,436.00 (99.19%) | 987.00 (98.70%) | |
| Yes | 82.00 (0.86%) | 69.00 (0.81%) | 13.00 (1.30%) | |
| CVD, n (p%) | | | | <0.001 |
| No | 8,409.00 (88.47%) | 7,592.00 (89.27%) | 817.00 (81.70%) | |
| Yes | 1,096.00 (11.53%) | 913.00 (10.73%) | 183.00 (18.30%) | |
| Smoke, n (p%) | | | | <0.001 |
| No | 5,783.00 (60.84%) | 5,118.00 (60.18%) | 665.00 (66.50%) | |
| Yes | 3,722.00 (39.16%) | 3,387.00 (39.82%) | 335.00 (33.50%) | |
| Drink, n (p%) | | | | 0.006 |
| No | 5,568.00 (58.58%) | 4,942.00 (58.11%) | 626.00 (62.60%) | |
| Yes | 3,937.00 (41.42%) | 3,563.00 (41.89%) | 374.00 (37.40%) | |
| BMI, n (p%) | | | | <0.001 |
| Normal | 625.00 (6.58%) | 549.00 (6.46%) | 76.00 (7.60%) | |
| Overweight | 7,796.00 (82.02%) | 7,018.00 (82.52%) | 778.00 (77.80%) | |
| Obese | 1,084.00 (11.40%) | 938.00 (11.03%) | 146.00 (14.60%) | |

Values are presented as mean±SD for continuous variables and as numbers (%) for categorical variables. *P*-values were based on t-tests or chi-square tests, as appropriate.

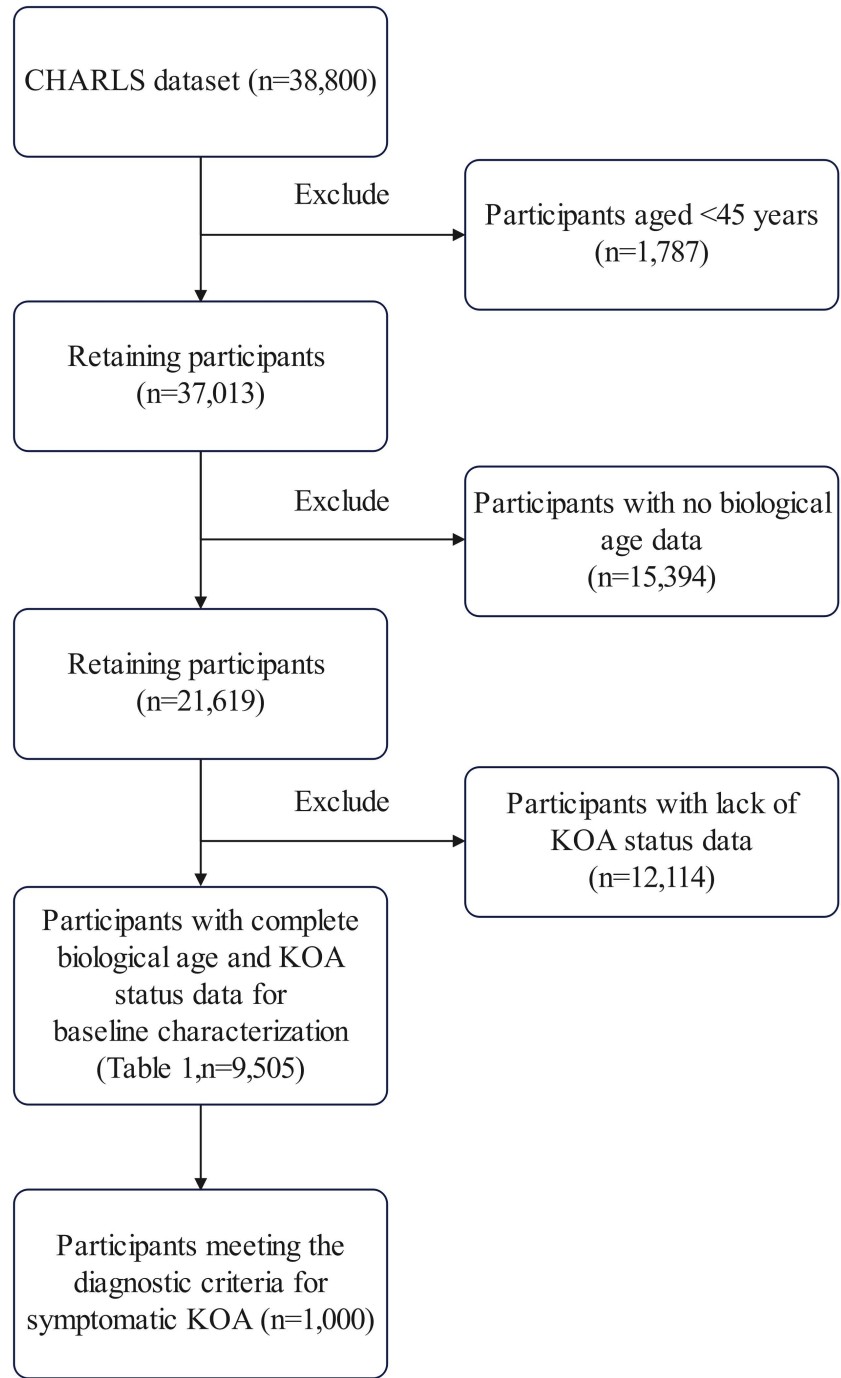

**Fig 1. Flowchart of participant selection.**

and machine learning models (e.g., lifestyle factors and comorbidities), which had a low rate of missingness (<2%), we employed multiple imputation by chained equations (MICE) to handle missing values and preserve the analytical sample size.

## Variable definitions

The outcome variable was symptomatic KOA, defined as self-reported physician-diagnosed osteoarthritis accompanied by knee joint pain, as ascertained from participant responses in the CHARLS survey. This definition, which is widely used in large-scale epidemiological studies because of its feasibility, lacks radiographic confirmation (e.g., Kellgren-Lawrence grading) and may be subject to misclassification bias. The primary exposure was BA, which was quantified using the KDM [17]. While the standard KDM algorithm incorporates 12 biomarkers, albumin, red blood cell count, ferritin, and transferrin were unavailable in CHARLS. Following the validation by Liu and Ruan, who confirmed the reliable replicability of the remaining 8 biomarkers for aging assessment in the Chinese population, the present study incorporated serum total cholesterol, triglycerides, glycosylated hemoglobin, urea nitrogen, creatinine, systolic blood pressure, high-sensitivity C-reactive protein, and platelet count, with reference to their calculation methods [18,19]. Log transformation was applied to the 8 serum biomarkers to approximate normal distribution. BA was analyzed as a continuous variable and categorized into quartiles. Covariates were selected based on the prior literature and included demographic (age, gender, marital status, education level, and residence), lifestyle (smoking and drinking), and clinical variables (hypertension, dyslipidemia, diabetes, cancer, cardiovascular disease, and BMI category). To enhance transparency regarding the input variables, the distributions (histograms) of the eight serum biomarkers used for BA calculation and the resulting BA itself are provided in Supplementary S1-S9 Fig, allowing for assessment of data characteristics such as skewness and outliers.

## Statistical analysis

**Descriptive statistics.** Baseline characteristics were summarized using means and standard deviations for continuous variables and frequencies and percentages for categorical variables. Between-group comparisons (KOA vs. non-KOA) were assessed using Welch's t-test for continuous variables and Pearson's chi-square test for categorical variables.

**Logistic regression models.** Multivariate logistic regression analysis was performed to assess the association between BA and symptomatic KOA. Three progressively adjusted models were used in this study: The first model included no covariate adjustments. The second model was adjusted for gender, education level, and residence. The fully adjusted model additionally controlled for BMI category, smoking status, alcohol consumption, hypertension, dyslipidemia, cardiovascular disease, cancer, and diabetes mellitus status. BA was examined as a continuous variable and quartile, with the lowest quartile serving as the reference. Trend tests across quartiles were conducted by modeling the ordinal quartile variables as continuous terms. Odds ratios (ORs) and 95% confidence intervals (CIs) were calculated for the analyses.

**Restricted Cubic Spline (RCS) analysis.** To explore potential non-linear associations, restricted cubic spline logistic regression was applied with three knots placed at the 10th, 50th, and 90th percentiles of BA. The median value was used as the reference point. Adjustments were made for all covariates in Model 3. The statistical significance of the nonlinear terms was assessed using likelihood ratio tests.

**Subgroup analyses.** Stratified logistic regression models were performed across subgroups defined by sex, residence, BMI category, smoking status, and presence of comorbidities (e.g., hypertension and cardiovascular disease). Each model was adjusted for all covariates, except the stratification variable. Interaction terms were used to test for heterogeneity between the subgroups.

**Machine learning models.** Six supervised classification models were constructed to assess the association between features and symptomatic KOA status: Decision Tree, Random Forest (RF), Light Gradient Boosting Machine (LightGBM), XGBoost, CatBoost, and support vector machine (SVM). The model performance was evaluated using accuracy, recall, specificity, F1-score, precision, Matthews correlation coefficient (MCC), and area under the receiver operating characteristic curve (AUROC). Both the hold-out test set and 5-fold cross-validation were used for model validation.

Feature selection was performed using LASSO regression, which identified nonzero coefficient factors for model inclusion. Prior to modeling, continuous variables (e.g., BA, BMI, and blood biomarkers) were standardized using z-scores (mean = 0, SD = 1), and categorical variables (e.g., gender and education) were one-hot encoded. Covariates

with minimal missingness (<2%) were handled using multiple imputation by chained equations (MICE). To mitigate overfitting given the sample size, the dataset was split into a 70% training set and a 30% testing set, stratified by KOA status. The model implementation details were as follows: Decision Tree and Random Forest used the Gini impurity criterion; gradient boosting models (XGBoost, LightGBM, CatBoost) were tuned for maximum depth, learning rate, and number of estimators, incorporating L1/L2 regularization and early stopping (50 rounds); and Support Vector Machine (SVM) employed a radial basis function (RBF) kernel with regularization parameter C tuned via five-fold cross-validation. These measures contributed to stable model performance, as evidenced by consistent AUROC values (>0.90) across the cross-validation folds.

**Model Interpretation with SHAP.** SHapley Additive exPlanations (SHAP) were used to interpret the outputs of the best-performing model (XGBoost). The SHAP values quantify the contribution of each feature to the model output. SHAP bar plots, beeswarm plots, force plots, and heatmaps were used to visualize the global and local interpretability.

Statistical significance was set at $p < 0.05$. All analyses were performed using R (version 4.4.3) and DecisionLinnc 1.0.

## Results

### Baseline characteristics of participants by KOA status

A total of 9,505 participants were included in the analysis, comprising 1,000 individuals with and 8,505 without symptomatic KOA. Compared with those without symptomatic KOA, individuals with symptomatic KOA were older (mean age: 60.87 vs. 59.37 years, $p < 0.001$) and had a higher BA (mean: 59.97 vs. 58.76 years, $p < 0.001$). There was no significant difference in BMI between the groups ($p = 0.122$). Participants with symptomatic KOA were more likely to be female, live in rural areas, have lower educational levels, and have a higher prevalence of hypertension, dyslipidemia, and cardiovascular disease (all $p < 0.05$). The detailed characteristics are listed in Table 1. The analytical sample for subsequent modeling comprised 1,000 individuals with complete covariate data available.

### Variable selection using lasso regression

The Lasso regression identified 11 non-zero coefficient factors at the optimal lambda value (lambda.min = 0.0005), including BA, gender, education, residence, hypertension, dyslipidemia, cardiovascular disease (CVD), smoking, drinking, BMI category, and cancer. Variables such as diabetes were excluded, with coefficients shrinking to zero, suggesting no substantial independent association with KOA in the penalized model.

As shown in the coefficient trajectory plot (Fig 2A), increasing the regularization parameter lambda led to progressive shrinkage of the coefficients, with some variables dropping as lambda increased. The cross-validation curve (Fig 2B) confirmed that the model achieved optimal performance at log(lambda) = −7.695 (lambda.min), with a stable mean cross-validated deviance.

These results suggest that BA remains an independent and stable variable of symptomatic KOA, even when penalized for multicollinearity and variable redundancy.

### Association between BA and symptomatic KOA

As shown in Table 2, a higher BA was significantly associated with an increased likelihood of developing symptomatic KOA. When modeled as a continuous variable, each one-year increase in BA was associated with a 1.23% to 1.58% increase in symptomatic KOA odds across all three models. In the fully adjusted model (Model 3), the odds ratio was 1.0123 (95% *CI*: 1.0049–1.0197, $p = 0.0010$).

A positive dose-response relationship was observed when BA was categorized into quartiles. Compared with the lowest quartile (Q1), participants in Q3 and Q4 had significantly higher odds of symptomatic KOA, with adjusted ORs of 1.4655 (95% *CI*: 1.1989–1.7940, $p = 0.0002$) and 1.4519 (95% *CI*: 1.1755–1.7956, $p = 0.0001$), respectively, in Model 3. The linear

A

### Lasso Regression Lambda and Coefficients Plot

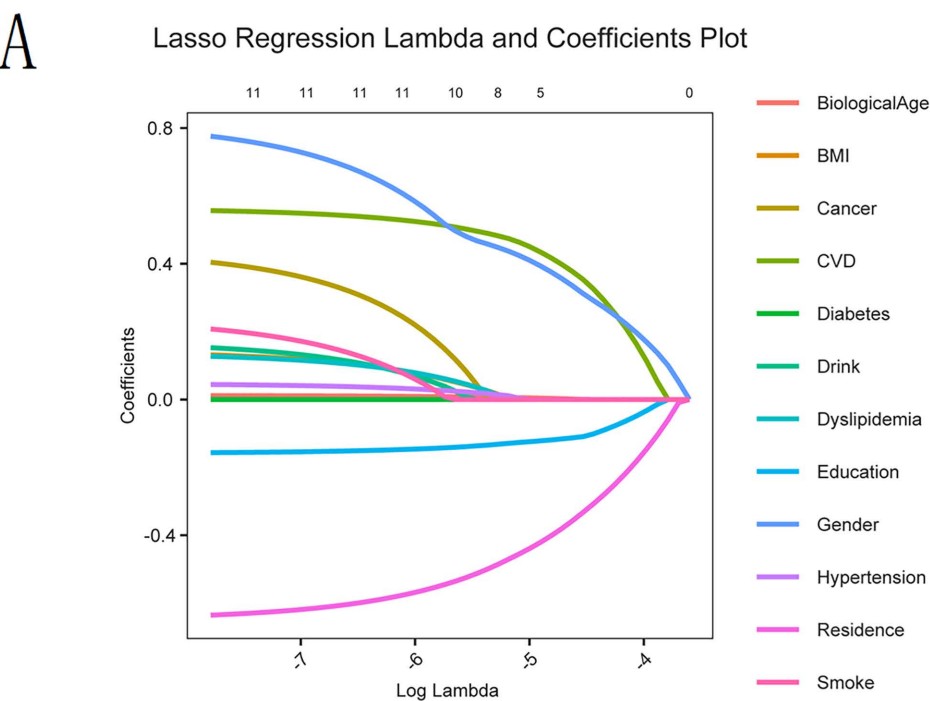

B

### Lasso Regression Lambda and CVM Plot

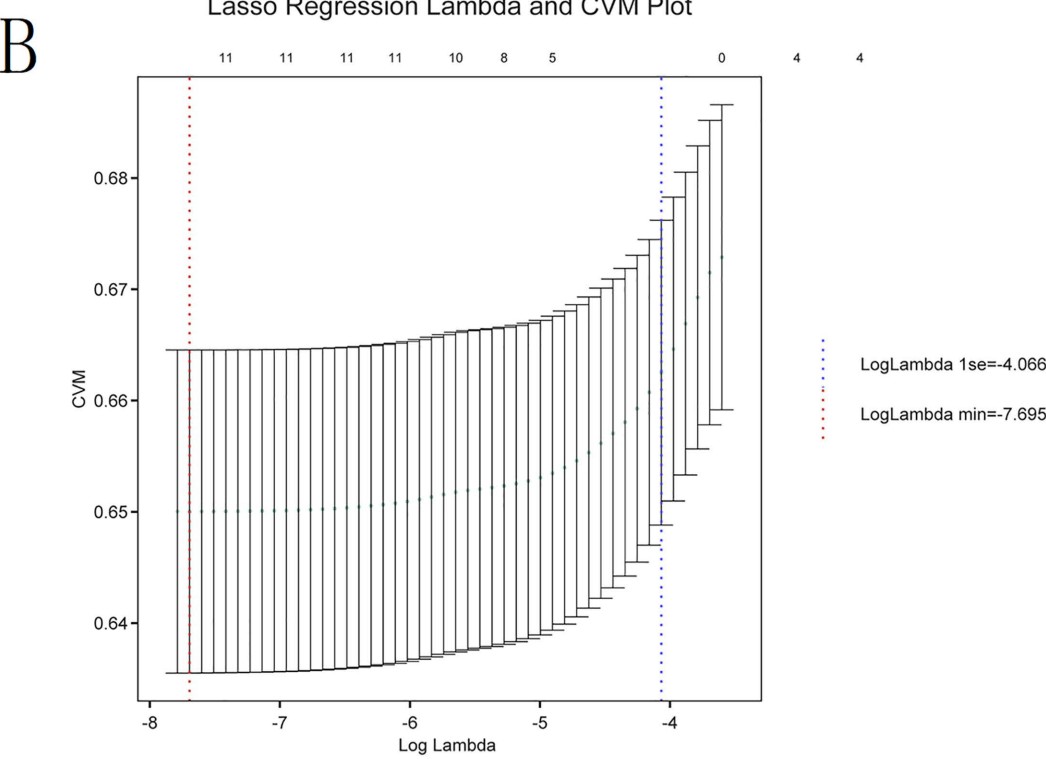

**Fig 2. LASSO coefficient path plot for symptomatic KOA feature selection.** (A) Coefficient paths for the LASSO logistic regression model. (B) Cross-validation curve for optimal lambda selection.

**Table 2. Association between BA and symptomatic KOA from logistic regression models.**

| Variable | OR (95% *CI*), *p*-value | | |
|---|---|---|---|
| | Model 1 | Model 2 | Model 3 |
| BA | **1.0121(1.0055,1.0187)0.0003** | **1.0158(1.0088,1.0228)<0.0001** | **1.0123(1.0049,1.0197)0.0010** |
| Stratified by BA quartiles | | | |
| Q1 | Reference | Reference | Reference |
| Q2 | 1.1813(0.9701,1.4397)0.0979 | 1.2679(1.0373,1.5510)0.0206 | 1.2064(0.9850,1.4788)0.0701 |
| Q3 | 1.4552(1.2040,1.7618)0.0001 | 1.5661(1.2875,1.9081)<0.0001 | 1.4655(1.1989,1.7940) 0.0002 |
| Q4 | 1.4370(1.1885,1.7404)0.0002 | 1.6013(1.3083,1.9629)<0.0001 | 1.4519(1.1755,1.7956) 0.0001 |
| *P* for trend | **<0.0001** | **<0.0001** | **0.0001** |

Model 1: Adjusted for none.

Model 2: Adjusted for gender, education, and residence.

Model 3: Adjusted for Model 2 combined with BMI, cancer, smoking, drinking, hypertension, dyslipidemia, and CVD.

trend across quartiles was statistically significant in all models (all *p* for trend<0.001), indicating a robust and graded association between increasing BA and the prevalence of symptomatic KOA.

### Nonlinear association between BA and symptomatic KOA

The RCS regression model demonstrated a statistically significant overall association between BA and symptomatic KOA (*p*<0.001) and a nonlinear relationship (*p*=0.013). As shown in Fig 3, the odds of symptomatic KOA gradually increased with BA, particularly beyond approximately 66.7 years. Below this threshold, the association appeared relatively flat, suggesting a nonlinear dose-response pattern.

The spline curve indicated that the risk of symptomatic KOA remained stable at lower BA levels but began to increase steeply beyond the inflection point, reflecting an accelerated risk in older individuals. This nonlinearity emphasizes that BA is not linearly associated with KOA risk across its entire range and supports its role as a sensitive biomarker for aging-related musculoskeletal conditions.

### Subgroup analyses

The association between BA and symptomatic KOA was consistent across the most predefined subgroups (Fig 4). Notably, the association was slightly stronger among females, rural residents, and those with CVD. For example, in participants with CVD, the OR per unit increase in BA was higher than that in participants without CVD, suggesting a potential synergistic effect between vascular burden and musculoskeletal aging.

However, no significant interactions were detected across strata (*p* for interaction>0.05 for all), indicating that the relationship between BA and symptomatic KOA was broadly stable and not meaningfully modified by the baseline characteristics.

These findings support the robustness of the observed association and suggest that BA is a consistent variable of symptomatic KOA risk across a range of population subgroups.

### Performance of machine learning models in discriminating symptomatic KOA

Among all the models evaluated on the testing set, the XGBoost model achieved the best overall performance (AUROC=0.9078, accuracy=83.37%, recall=87.98%, and F1-score=85.39%). The LGBM model also demonstrated excellent performance (AUROC=0.8973, F1-score=84.03%), followed by CatBoost (AUROC=0.8616) and Random Forest (AUROC=0.7393).

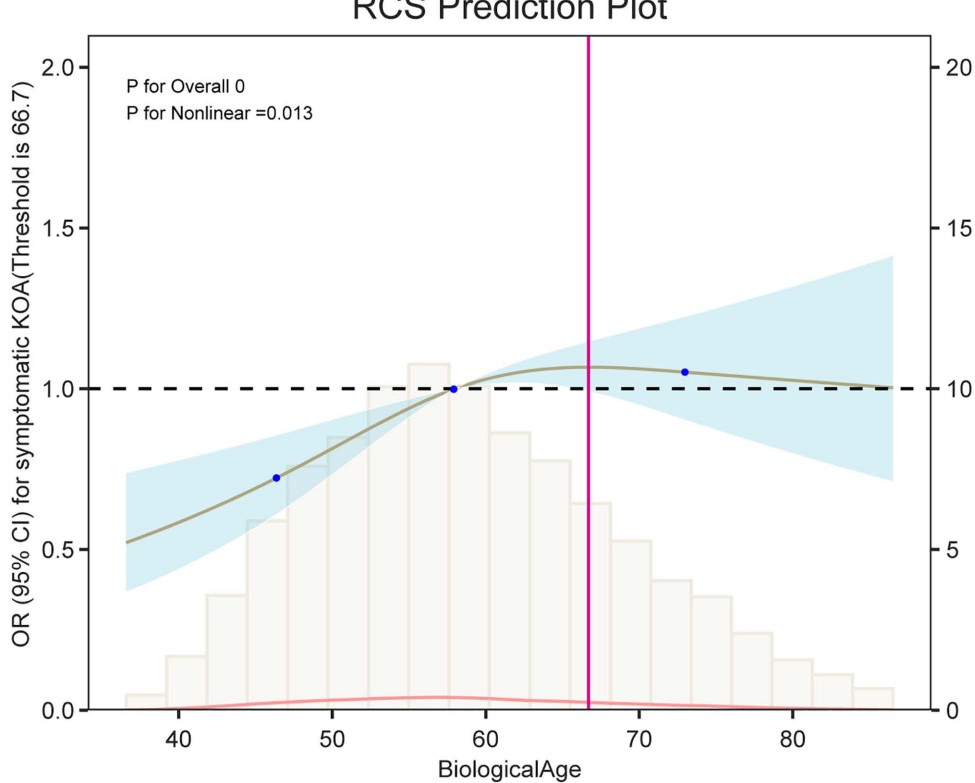

**Fig 3. The RCS analysis showed a non-linear association between BA and symptomatic KOA risk.** The solid line represents the adjusted odds ratios (ORs), and the shaded area denotes the 95% confidence intervals (CIs) derived from the RCS regression with three knots at the 10th, 50th, and 90th percentiles. The reference point was set at a median BA of 66.7 years. A significant non-linear association was observed ($p$ for overall association < 0.001; $p$ for non-linearity = 0.013). The risk of symptomatic KOA increased markedly when BA exceeded approximately 66.7 years, indicating a critical threshold for risk escalation.

In contrast, the SVM and Decision Tree models showed relatively lower discriminative abilities, with AUROCs of 0.7159 and 0.6965, respectively. The mean AUROC across all models was 0.8031. A summary of the test set performance metrics is presented in Table 3, and the corresponding ROC curves are shown in Fig 5.

These results indicate that gradient boosting-based models, particularly XGBoost, are more suitable for distinguishing individuals with symptomatic KOA based on BA and clinical features than other models.

### Cross-validation performance of XGBoost model

The robustness of the XGBoost model was further confirmed through 5-fold cross-validation, which yielded consistently high performance across all folds, with a mean AUROC of 0.9106, mean accuracy of 83.68%, and mean F1-score of 85.58% (Table 4). The corresponding ROC curves for each fold are displayed in Fig 6, all demonstrating strong discriminatory power (AUROC > 0.90). A comprehensive set of performance metrics for each individual fold is available in Supplementary S1 Table.

### Feature contributions in the XGBoost model

SHAP analysis identified BA as the most influential feature in the XGBoost model for symptomatic KOA patients (Fig 7A). In the SHAP bar plot, the mean absolute SHAP value, represented by the length of the bars, quantifies the average

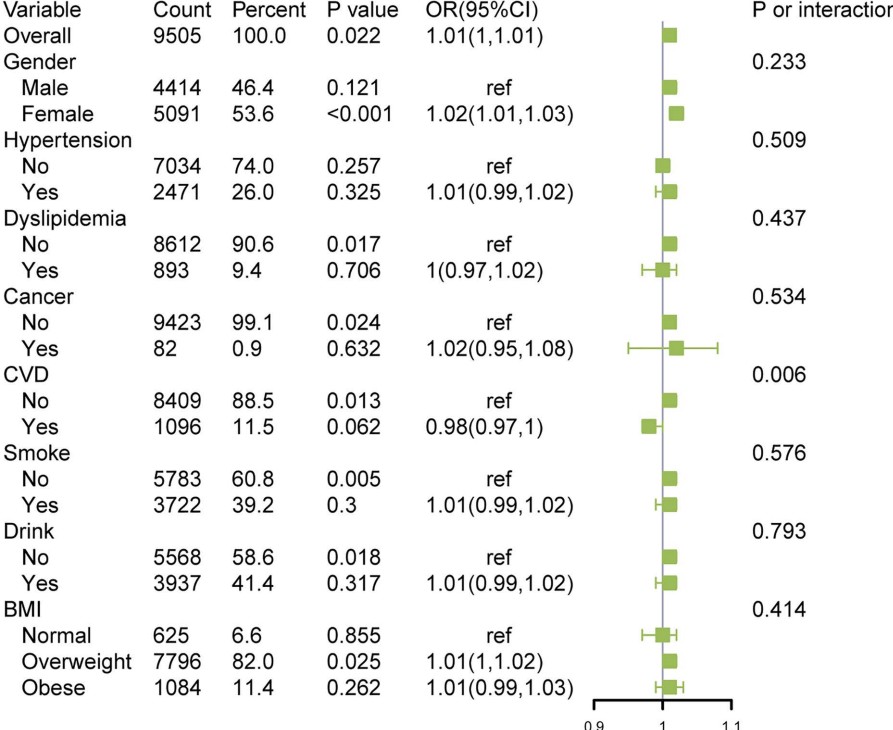

Univariate Subgroup GLM Regression Forest Plot

| Variable | Count | Percent | P value | OR(95%CI) | P or interaction |
|---|---|---|---|---|---|
| Overall | 9505 | 100.0 | 0.022 | 1.01(1,1.01) | |
| Gender | | | | | 0.233 |
| Male | 4414 | 46.4 | 0.121 | ref | |
| Female | 5091 | 53.6 | <0.001 | 1.02(1.01,1.03) | |
| Hypertension | | | | | 0.509 |
| No | 7034 | 74.0 | 0.257 | ref | |
| Yes | 2471 | 26.0 | 0.325 | 1.01(0.99,1.02) | |
| Dyslipidemia | | | | | 0.437 |
| No | 8612 | 90.6 | 0.017 | ref | |
| Yes | 893 | 9.4 | 0.706 | 1(0.97,1.02) | |
| Cancer | | | | | 0.534 |
| No | 9423 | 99.1 | 0.024 | ref | |
| Yes | 82 | 0.9 | 0.632 | 1.02(0.95,1.08) | |
| CVD | | | | | 0.006 |
| No | 8409 | 88.5 | 0.013 | ref | |
| Yes | 1096 | 11.5 | 0.062 | 0.98(0.97,1) | |
| Smoke | | | | | 0.576 |
| No | 5783 | 60.8 | 0.005 | ref | |
| Yes | 3722 | 39.2 | 0.3 | 1.01(0.99,1.02) | |
| Drink | | | | | 0.793 |
| No | 5568 | 58.6 | 0.018 | ref | |
| Yes | 3937 | 41.4 | 0.317 | 1.01(0.99,1.02) | |
| BMI | | | | | 0.414 |
| Normal | 625 | 6.6 | 0.855 | ref | |
| Overweight | 7796 | 82.0 | 0.025 | 1.01(1,1.02) | |
| Obese | 1084 | 11.4 | 0.262 | 1.01(0.99,1.03) | |

**Fig 4. Subgroup analysis of the association between BA and symptomatic KOA.** Forest plot displaying the odds ratios (95% CIs) for the association between BA and symptomatic KOA across the prespecified subgroups. No significant interactions were observed, indicating consistent effects across the strata.

**Table 3. Performance metrics of machine learning models (test set).**

| ModelName | Accuracy | Recall | F1-Score | MCC | AUROC | Precision | Specificity | FNR | FPR |
|---|---|---|---|---|---|---|---|---|---|
| DecisionTreeTEST | 0.6788 | 0.8388 | 0.7426 | 0.3460 | 0.6965 | 0.6662 | 0.4815 | 0.1612 | 0.5185 |
| LGBMTEST | 0.8174 | 0.8699 | 0.8403 | 0.6296 | 0.8973 | 0.8127 | 0.7525 | 0.1301 | 0.2475 |
| RFTEST | 0.6800 | 0.8377 | 0.7431 | 0.3484 | 0.7393 | 0.6677 | 0.4855 | 0.1623 | 0.5145 |
| CatBoostTEST | 0.7812 | 0.8585 | 0.8125 | 0.5561 | 0.8616 | 0.7712 | 0.6858 | 0.1415 | 0.3142 |
| XGBTEST | 0.8337 | 0.8798 | 0.8539 | 0.6628 | 0.9078 | 0.8295 | 0.7768 | 0.1202 | 0.2232 |
| SVMTEST | 0.6499 | 0.7836 | 0.7120 | 0.2825 | 0.7159 | 0.6524 | 0.4848 | 0.2164 | 0.5152 |

The metrics included accuracy, F1-score, recall, specificity, Matthews correlation coefficient (MCC), and AUROC. XGBoost outperformed all models across most indicators.

impact of each feature on the model output. BA exhibited the longest bar (value >0.6), confirming its dominant role, while other contributing features (residence, gender, education, marital status, cardiovascular disease, BMI category, and hypertension) showed shorter bars, indicating weaker albeit non-negligible effects.

Fig 7B (SHAP beeswarm plot) further elucidates the relationships between features and KOA risk. Color intensity reflects the feature values, with purple indicating high values and yellow indicating low values (e.g., purple points correspond to high BA). The horizontal axis (SHAP value) indicates the direction of the feature's contribution: positive values

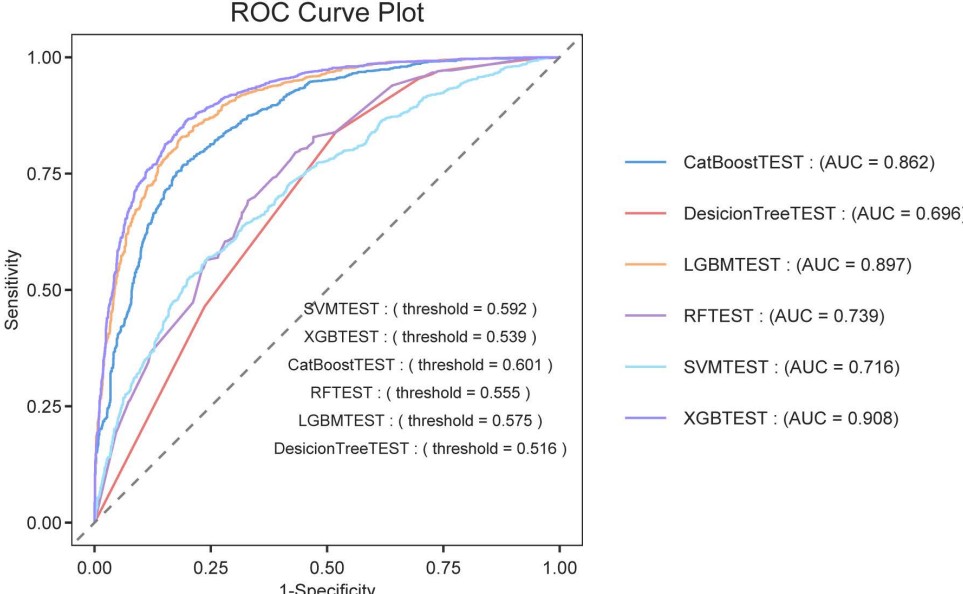

**Fig 5. ROC curves of the six machine learning models for distinguishing symptomatic KOA (test set).** The curves compare the discriminatory performance of the six classifiers (XGBoost, LGBM, CatBoost, RF, Decision Tree, and SVM) for identifying individuals with symptomatic KOA. The XGBoost model exhibited the highest AUROC (0.9078), indicating its superior performance.

**Table 4. Cross-validation results of the XGBoost model.**

| ModelName | Accuracy | F1-Score | AUROC |
|---|---|---|---|
| XGB_1TEST | 0.8438 | 0.8602 | 0.9154 |
| XGB_2TEST | 0.8383 | 0.8569 | 0.9169 |
| XGB_3TEST | 0.8261 | 0.846 | 0.9022 |
| XGB_4TEST | 0.8424 | 0.8605 | 0.9082 |
| XGB_5TEST | 0.8333 | 0.8555 | 0.91 |
| mean_scores | 0.8368 | 0.8558 | 0.9106 |

are associated with an increase in the model output value for symptomatic KOA, whereas negative values are associated with a decrease. Higher BA values (purple points) were consistently associated with positive SHAP values. The horizontal spread of points reflects the variation in influence across individuals, and the wide dispersion of BA points indicates its consistently positive yet heterogeneous effect. Other features, such as rural residence, also clustered predominantly within positive SHAP values, suggesting an association with an elevated model-estimated risk.

Individual-level explanations derived from SHAP force and waterfall plots (Fig 8) illustrated how combinations of clinical and demographic features collectively determined the model output. Additionally, the SHAP heatmap and line plots (Fig 9) revealed consistent patterns across individuals, reinforcing the role of BA and related features as key determinants of the XGBoost model output.

## Discussion

This cross-sectional analysis of the CHARLS dataset revealed a robust association between BA and symptomatic KOA prevalence. Each 1-year increase in BA was associated with 1.23%–1.58% higher odds of KOA (fully adjusted *OR* =

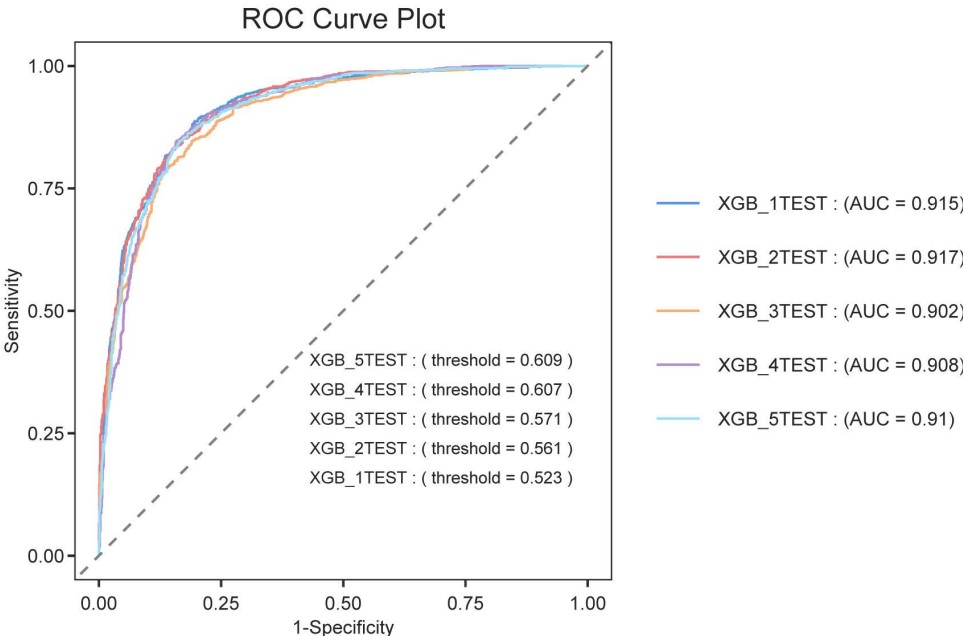

**Fig 6. ROC curves from the five-fold cross-validation of the XGBoost model.** The curves depict the model's performance in distinguishing cases of symptomatic KOA across all cross-validation folds. All folds demonstrated strong and consistent discriminatory performance, with AUROC > 0.90.

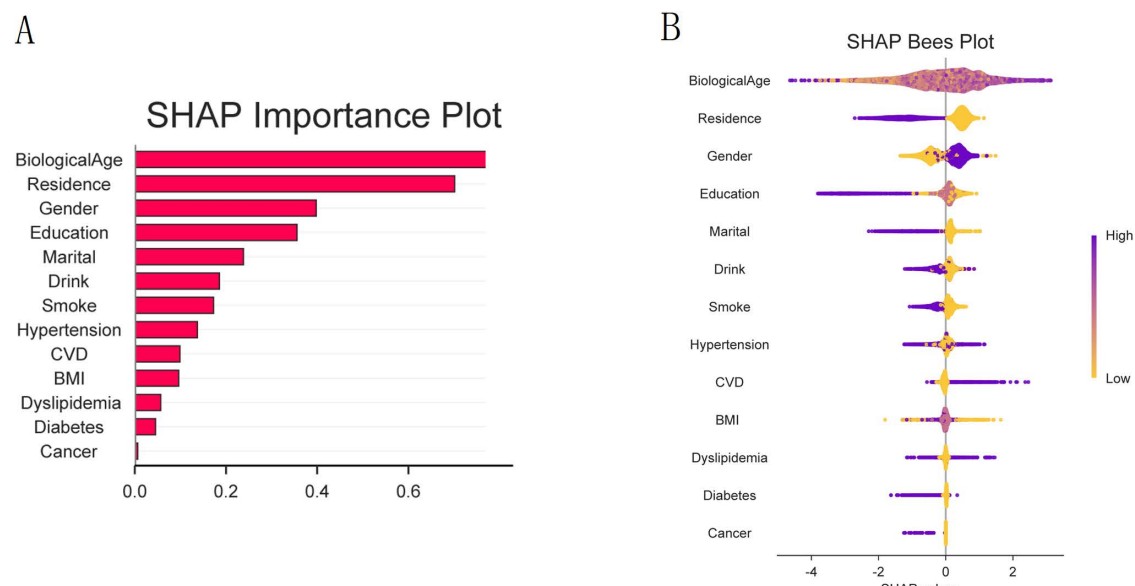

**Fig 7. Analysis of feature associations and contributions of the XGBoost model.** (A) SHAP bar plot of the mean absolute SHAP values. The length of each bar represents the average magnitude of a feature's contribution to the model's output. BA made the largest average contribution, followed by residence, gender, and education. (B) SHAP beeswarm plot showing the distribution of each feature's contribution. Each point represents an individual participant. The color indicates the feature value (purple indicates high and yellow indicates low). The horizontal position (SHAP value) shows the direction and magnitude of the feature's contribution to the model output for that individual (positive values are associated with a higher model output for symptomatic KOA).

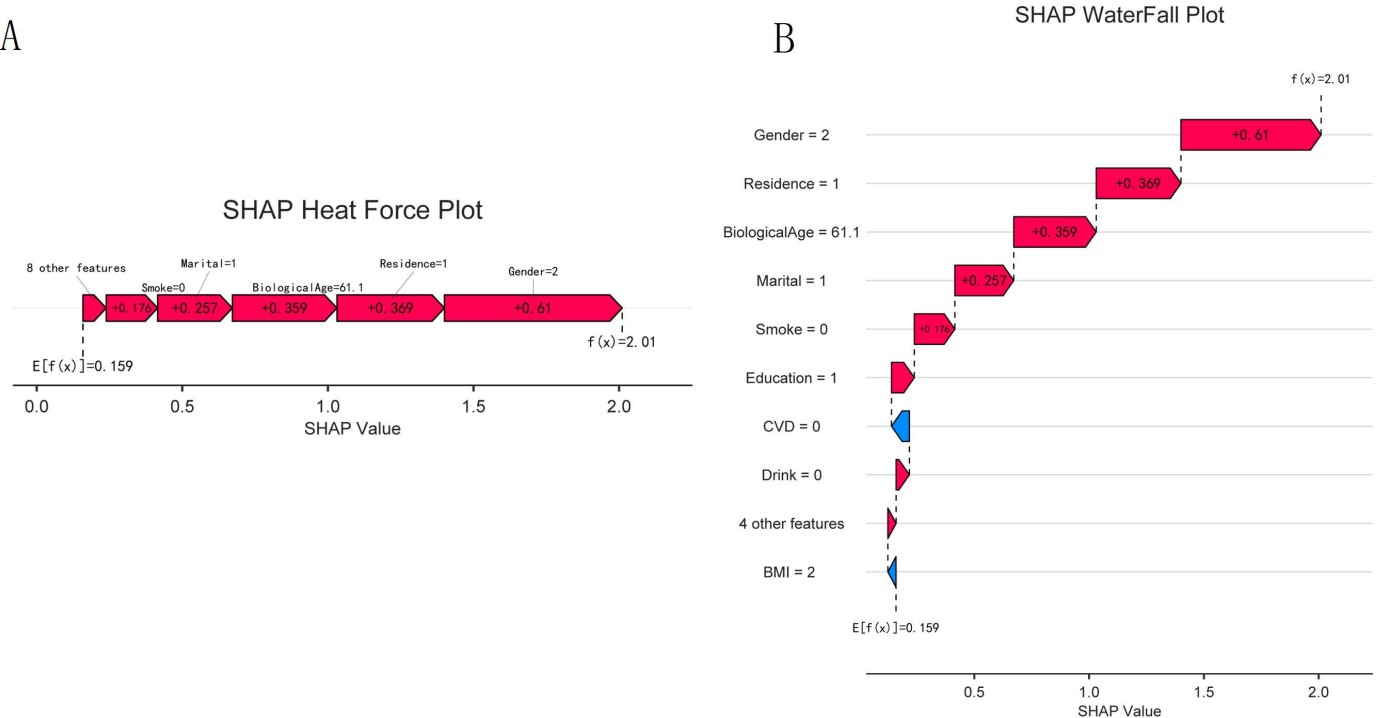

**Fig 8. Individual-level explanation of the XGBoost model output using SHAP.** (A) SHAP force plot for a single participant. The features that pushed the model output value away from the base value (the average model output) are shown. The features on the right are associated with an increase in the output value, whereas those on the left are associated with a decrease. (B) SHAP waterfall plot for the same participant, illustrating the cumulative contribution of each feature to the final model output. These plots exemplify how the model integrates multiple factors to arrive at an output for an individual, with BA being the primary contributing feature in this case.

1.0123, 95% *CI*: 1.0049–1.0197, *p* = 0.001), with a graded dose-response relationship. Participants in the highest BA quartile had a 45.19% higher risk of symptomatic KOA than those in the lowest quartile (*OR* = 1.4519, 95% *CI*: 1.1755–1.7956). Restricted cubic spline analysis revealed a non-linear association, with symptomatic KOA risk accelerating steeply beyond BA ≥ 66.7 years (non-linearity *p* = 0.013). These findings are consistent with international evidence from two cross-sectional studies in the U.S. National Health and Nutrition Examination Survey (NHANES) [20,21]. Both studies consistently demonstrated positive associations between KDM-based biological aging indicators (including BA acceleration) and the risk of osteoarthritis. In the fully adjusted models, the highest quartile of BA acceleration was associated with a 36.3% increase in OA risk (*OR* = 1.363, 95% *CI*: 1.213–1.532), with evident dose-response relationships. Additionally, sex-specific effects were observed, with stronger associations identified in females than in males. One study further revealed that biological aging mediates the relationship between central obesity and OA, with a mediation proportion ranging from 0% to 40%. Together, these results support the cross-population robustness of KDM-based biological aging measures as reliable markers for OA risk stratification. The XGBoost machine learning model achieved an accuracy of 83.37% and an AUROC of 0.9078, with SHAP values confirming BA as the most influential feature (mean absolute SHAP value >0.6). These findings underscore the importance of BA as a pivotal biomarker for the risk stratification of symptomatic KOA.

Furthermore, the divergence between BA and CA underscores heterogeneous aging trajectories, likely reflecting cumulative exposure to environmental, lifestyle, and genetic factors. The accelerated risk of symptomatic KOA beyond a BA threshold of ~66.7 years suggests that individuals with advanced biological aging may experience a compressed health

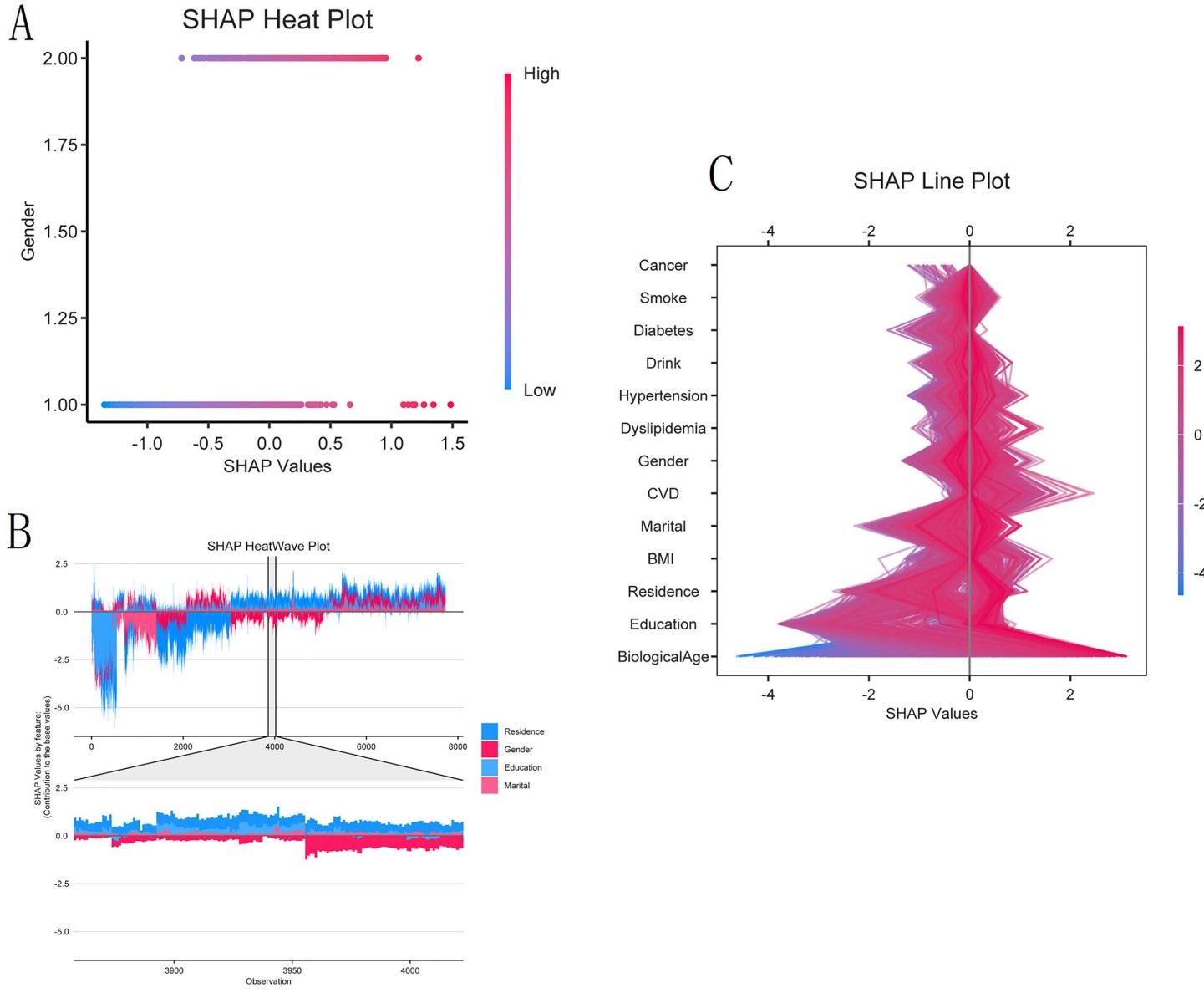

**Fig 9. SHAP heatmaps and line plots depicting feature-influence dynamics in symptomatic KOA.** (A) SHAP heatmap of feature associations in the cohort. This plot visualizes the variation in the contribution of each feature to the model output for all individuals. Warmer colors indicate a greater magnitude of the contribution. (B)SHAP heatwave plot of the feature contribution trajectories. Sequential SHAP values by feature and observation index showing the dynamic influence of each variable across the dataset.(C) SHAP line plot of the feature contributions. The plot shows the SHAP values for the most influential features across all individuals in the cohort. Each line represents a feature, and its vertical position and density indicate the direction and magnitude of its association with the model output for the symptomatic KOA. BA exhibited the strongest and most consistent positive association.

span, with an earlier onset of age-related disorders. Future longitudinal studies are needed to explore whether BA acceleration is associated with overall lifespan reduction or earlier disability.

Our results align closely with those of existing mechanistic and epidemiological studies, enhancing our understanding of the relationship between BA and symptomatic KOA. Mechanistically, previous studies have established that cellular senescence and the subsequent SASP play pivotal roles in pathogenesis [22,23]. At the cellular level, senescent

chondrocytes within the articular cartilage exhibit a decline in matrix synthesis and upregulation of catabolic factors such as matrix metalloproteinases (MMPs) and pro-inflammatory cytokines such as interleukin-6 (IL-6) and tumor necrosis factor-alpha (TNF-α) [24–26]. These factors disrupt the normal balance between cartilage synthesis and degradation, leading to progressive cartilage erosion and joint inflammation and accelerating the process of KOA [27–29].

In the context of biological aging, systemic factors associated with BA acceleration likely contribute to local joint pathology. For instance, elevated oxidative stress, a common feature of increased BA, can directly damage chondrocytes and extracellular matrix components [30,31]. It can also promote the activation of nuclear factor kappa B (NF-κB), a key transcription factor that upregulates the production of SASP factors, thus creating a pro-inflammatory microenvironment conducive to KOA development [32,33]. Additionally, chronic low-grade inflammation associated with aging, as reflected by elevated BA levels, may dysregulate immune cell function in joints, including macrophages and T lymphocytes [34,35]. These immune cells secrete cytokines and proteases, exacerbating cartilage damage and synovial inflammation [36–38].

Epidemiologically, our study extends previous work by demonstrating a population-level association between BA and symptomatic KOA using a large, nationally representative Chinese cohort, addressing the limitation that many prior epidemiological investigations predominantly relied on CA as a risk factor for symptomatic KOA while overlooking the potential value of BA, a metric that integrates multiple biomarkers reflecting an individual's overall biological state. By incorporating BA into the analysis, we provided a more nuanced understanding of disease risk, with the identified non-linear BA-symptomatic KOA relationship supporting the concept proposed in recent literature of critical biological aging thresholds beyond which the risk of joint degenerative diseases, such as KOA, may increase exponentially, consistent with earlier research highlighting BA as a powerful feature of various age-related pathologies, including musculoskeletal disorders [39–41]. In terms of discriminative accuracy, BA's multisystem biomarker panel offers a more comprehensive measure of an individual's functional decline compared to CA, as further validated by our machine learning results using the XGBoost model, wherein BA served as a top feature with high accuracy and AUROC values, aligning with emerging trends advocating composite biomarkers to improve the assessment of age-related disease risk.

Importantly, our findings demonstrated the incremental value of BA beyond established risk factors such as CA and BMI. This study advances the conventional risk prediction model developed by Wang et al., which relies on demographic variables including age and sex (AUC = 0.719) [4], by introducing BA as an innovative indicator. BA not only integrates biomarkers reflective of multi-system biological aging, but also, when incorporated into the XGBoost model, exhibits the potential to enhance predictive performance, with an achieved AUROC of 0.9078, thereby providing a new dimension for more precise stratification of high-risk populations. While CA is well recognized, BA captures biological heterogeneity within age groups and is strongly associated with symptomatic KOA risk, supported by three key observations: BA emerged as the most influential feature in the robust XGBoost model, outperforming traditional factors in SHAP value contributions; its significant association with symptomatic KOA persisted after rigorous adjustment for covariates in logistic regression; and a BA-specific non-linear risk acceleration point was identified, which CA alone may obscure. This enhanced risk stratification has substantial clinical promise; However, the translation of BA assessment into routine clinical practice faces challenges, including the cost and feasibility of measuring multiple biomarkers, the need for population-specific validation of BA algorithms, and the generalizability of the findings to diverse healthcare settings. The strong association between BA and symptomatic KOA, coupled with its superior performance in discriminative models, suggests that BA may hold potential for enhancing risk stratification frameworks in future studies. Integrating BA assessment into clinical practice could refine screening protocols by identifying individuals with elevated symptomatic KOA risk attributable to their biological aging trajectory, even those with "lower-risk" CA or BMI. This facilitates targeted preventive interventions, including lifestyle modifications, enhanced monitoring, and future senotherapeutics, thereby maximizing benefits and ultimately delaying the onset or slowing the progression of symptomatic KOA. The current study had several limitations that merit consideration. First, its cross-sectional nature prevents definitive inference regarding causality between BA and symptomatic KOA, as temporal relationships cannot be established. Second, the diagnosis of

symptomatic KOA was based solely on self-reported clinical information without radiographic confirmation (e.g., Kellgren-Lawrence grading), which may have introduced misclassification bias. Future studies incorporating imaging assessments will strengthen diagnostic precision. Third, the BA calculation relied on a reduced set of 8 systemic biomarkers instead of the standard 12 used in the KDM method because of data availability constraints in the CHARLS. Although prior studies have validated similar reduced sets in Chinese populations [18,19], this approach may introduce bias and affect the absolute value of BA estimates, potentially limiting direct comparability with studies using the full biomarker panel. Future research should aim to utilize the complete set of biomarkers where possible. Fourth, the multistage participant selection process (Fig 1) necessary to obtain a sample with complete data for the core variables (BA and symptomatic KOA status) may have introduced selection bias. A direct sensitivity analysis was not feasible because of the lack of core variables in the excluded groups. Although the final analytical sample retained the broad characteristics of the CHARLS cohort, this potential bias may have affected the representativeness of our sample and limited the generalizability of our findings. Fifth, although occupational physical load and activity levels are potential contributors to KOA risk, we did not adjust for them in our primary model. This decision was based on the limitations of the available data (e.g., lack of detailed lifetime occupational history and objective physical activity measures) and the substantial risk of reverse causality, as prevalent KOA severely impacts an individual's current activities and work capacity. Future prospective studies with dedicated pre-symptomatic exposure assessments are needed to disentangle these complex relationships. Furthermore, although we performed rigorous internal validation through cross-validation and subgroup analyses, the lack of an independent external dataset for validation means that the generalizability of the observed associations may be limited to the study population. Future studies in larger, multicenter cohorts and diverse populations are needed to replicate these associations and confirm the robustness of BA as a marker for symptomatic KOA risk assessment. Finally, the generalizability of the findings to other ethnicities, younger populations, or clinical settings requires further validation, as this study focused on middle-aged and older Chinese adults from a community-based cohort. This study provides a basis for subsequent research. Longitudinal investigations are warranted to explore the dynamic interplay between BA trajectories and symptomatic KOA incidence or progression, which may help to elucidate potential causal relationships. Further studies integrating joint-specific aging biomarkers with systemic BA metrics could help refine risk stratification models for symptomatic KOA. Additionally, interventional studies examining whether targeting biological aging pathways (e.g., senolytics and anti-inflammatory agents) might mitigate symptomatic KOA risk could provide valuable insights into preventive strategies. Finally, refining machine learning models with multi-omics data (e.g., genomics and metabolomics) may enhance the discriminative utility of BA for symptomatic KOA, facilitating more personalized approaches to musculoskeletal health management.

## Conclusion

This study demonstrated that BA was significantly associated with symptomatic KOA in a large, nationally representative Chinese cohort, with a non-linear relationship characterized by an accelerated risk escalation beyond the critical BA threshold. The strong association between BA and symptomatic KOA, coupled with its superior performance in machine learning models, highlights its strong association with symptomatic KOA and its value for risk assessment. For instance, BA assessment may have the potential for integration into health screenings to better understand heterogeneous aging-related risk profiles for which targeted preventive strategies, such as lifestyle interventions or more frequent monitoring, could be initiated before the onset of severe symptoms. These findings underscore the utility of integrating BA into epidemiological investigations of age-related musculoskeletal disorders, potentially offering new insights into public health strategies aimed at the early identification and targeted prevention of symptomatic KOA. However, the cross-sectional design precludes causal inference, and reliance on self-reported KOA data without imaging confirmation is a limitation. Future longitudinal studies incorporating objective OA measures are needed to further elucidate the interplay between biological aging and symptomatic KOA pathogenesis.

## Supporting information

**S1 Fig. Distribution of C-reactive Protein (CRP).**
(TIF)

**S2 Fig. Distribution of Systolic Blood Pressure (SBP).**
(TIF)

**S3 Fig. Distribution of Glycated Hemoglobin (HbA1c).**
(TIF)

**S4 Fig. Distribution of Total Cholesterol.**
(TIF)

**S5 Fig. Distribution of Triglycerides.**
(TIF)

**S6 Fig. Distribution of Urea Nitrogen.**
(TIF)

**S7 Fig. Distribution of Creatinine.**
(TIF)

**S8 Fig. Distribution of Platelet Count.**
(TIF)

**S9 Fig. Distribution of Calculated Biological Age.**
(TIF)

**S1 Table. Detailed 5-fold cross-validation performance metrics of the XGBoost model.**
(XLSX)

**S1 Data. Raw data. Contains the de-identified dataset used for biological age calculation and symptomatic knee osteoarthritis analysis.**
(XLSX)

## Author contributions

**Data curation:** Peng Liu, Yufeng Wang.

**Formal analysis:** Yunli Wang.

**Funding acquisition:** Peng Liu, Yufeng Wang.

**Investigation:** Chang Liu.

**Project administration:** Peng Liu, Yufeng Wang.

**Resources:** Jiwei Lian.

**Software:** Jiwei Lian.

**Validation:** Yunli Wang.

**Visualization:** Chang Liu, Tingting Pang.

**Writing – original draft:** Fanyu Fu.

**Writing – review & editing:** Li Dong.

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
