## [Decision Letter · Decision Letter 0]

2 Sep 2025

Dear Dr. Wang,

Thank you for submitting your manuscript to PLOS ONE. After careful consideration, we feel that it has merit but does not fully meet PLOS ONE’s publication criteria as it currently stands. Therefore, we invite you to submit a revised version of the manuscript that addresses the points raised during the review process.

We look forward to receiving your revised manuscript.

Kind regards,

Ramada Rateb Khasawneh

Academic Editor

PLOS ONE

 [This study was supported by grants from the Sanming ProjectMedicine in Shenzhen(No.SZZYSM202402015) and Science and Technology Development Project of Jilin Province, China (No.20240304086SF)].

4. In the online submission form, you indicated that [All data were downloaded from the official website (https://charls.pku.edu.cn/index.htm) upon requests].

5. Please include a separate caption for each figure in your manuscript

Additional Editor Comments:

Reviewer #1:

The authors of the publication analyzed data from a relatively large group of patients and took great care to illustrate the statistical results. The statistical analysis methodology itself is rich and uses advanced methods.

A limitation of this analysis may be the lack of consideration of physical activity and work performed in the study population.

Another interesting aspect is the possible difference between biological and chronological age in the study population, as well as the reference to average life expectancy.

Reviewer #2:

总结� 作者使用 CHARLS 队列的数据调查了生物年龄 �BA� 与有症状的膝骨关节炎 �KOA� 之间的关联。他们应用逻辑回归、受限三次样条和机器学习模型�XGBoost、LightGBM 等来探索 BA 的预测价值。该研究报告了一种非线性关联在大约 66.7 年的 BA 后�KOA 风险加速。XGBoost 取得了最佳的预测性能�SHAP 确认 BA 是最有影响力的因素。作者得出结论�BA 是 KOA 风险分层的一种有前途的生物标志物。1. 学习规划横断面设计不允许因果推断但部分结论表明了预测或因果影响。建议重新构建主张以强调“关联”而不是“预测/因果关系”。样本选择和代表性在~38,800名参与者中只剩下9,505人最后分析了1,000例KOA病例。选择过程可能会引入偏差。建议详细解释排除标准、缺失数据处理和敏感性分析。2. KOA诊断标准有症状的 KOA 仅通过医生自我报告的疼痛诊断来定义缺乏影像学确认例如 KL 分级�。建议更详细地讨论错误分类的风险并在可能的情况下根据其他诊断定义验证结果。3. 生物年龄计算仅使用了 8 种生物标志物而不是 KDM-BA 中的标准 12 种。尽管通过事先验证证明了这一点但潜在的偏差并未得到充分解决。建议提供比较或敏感性分析并更明确地承认局限性。4. 机器学习方法对于复杂模型样本量�1,000 个 KOA 案例可能会受到限制从而引起过度拟合的担忧。没有使用外部验证数据集。建议为模型选择添加更有力的理由扩展过度拟合预防并阐明跨子集的性能是否稳健。5. 讨论与解释讨论夸大了 BA 的临床适用性而没有充分考虑成本、可行性和普遍性。建议缓和主张并包括与国际研究的比较。6. 小问题摘要和结论中的一些术语例如�“预测”、“因果关系”�应替换为更谨慎的措辞。澄清每个逻辑回归模型表中包含的协变量以提高透明度。数字例如�RCS、SHAP 图应包括更清晰的轴标签和临床解释。数据可用性声明应指定使用的确切 CHARLS 数据集版本。 该手稿通过创新地使用 BA 和机器学习解决了一个重要而新颖的问题。然而需要进行大量修订以解决方法论透明度、诊断准确性和研究结果的解释问题。我建议在重新考虑之前进行重大修改。

Reviewer #3:

This manuscript explores the association between biological age (BA) and symptomatic knee osteoarthritis (KOA) in Chinese adults using nationally representative CHARLS data, incorporating both traditional statistical modeling and machine learning approaches. The topic is timely and relevant, the data source is authoritative, and the modeling framework is generally sound. The finding of a nonlinear threshold relationship between BA and KOA risk is of potential epidemiological and public health significance. However, the manuscript still has several important areas for improvement in terms of theoretical framing, methodological transparency, terminology consistency, and result presentation. Therefore, I recommend major revision before the manuscript can be considered for publication in this journal.

1.Strengthen international comparison in the literature review

The introduction is mostly focused on the Chinese population and CHARLS data. While this is aligned with the study's aims, it limits international contextualization. I suggest including a brief overview of international research on the relationship between biological age and KOA—especially regarding differences in BA construction methods and clinical applications—to enhance the theoretical richness and global relevance of the study.

2.Clarify model structure and data preprocessing in the machine learning section

Although the manuscript evaluates six machine learning models, there is limited detail regarding model construction. Please clarify how features were selected, whether feature scaling or normalization was applied, how missing data were handled, and how overfitting was prevented. These additions would enhance reproducibility and model transparency.

3.Add a brief discussion of potential clinical applications of BA

While the predictive value of BA is emphasized, the practical implications are underdeveloped. I recommend briefly discussing how BA could potentially be integrated into clinical KOA screening or aging-related health assessments, such as early warning systems or personalized intervention strategies, to enhance the manuscript’s translational value.

4.Unify terminology and maintain consistency throughout the manuscript

The manuscript inconsistently uses terms like “biological age,” “BA,” and “KDM-BA.” I suggest defining the term clearly at first mention and consistently using “BA (KDM-based)” or another unified format throughout the paper to improve clarity and professionalism.

5.Provide clearer explanations for SHAP interpretation plots

Although SHAP plots are visually compelling, the textual explanations are brief. I suggest elaborating on what the colors (high/low feature values), directions (positive/negative effects), and spread (magnitude of influence) mean in the context of SHAP bar and beeswarm plots. This would help readers better interpret model results.

6.Add supplementary plots of key biomarker distributions

To increase transparency and interpretability, I suggest including histograms or boxplots of key input variables (e.g., CRP, SBP, HbA1c, etc.) and BA itself in the supplementary materials. This would allow readers to assess data skewness, outliers, and distribution assumptions.

7.Streamline the machine learning result presentation

The current machine learning results section is somewhat overloaded with detailed fold-level performance metrics. I suggest moving some of the five-fold cross-validation details (e.g., individual fold metrics) to supplementary materials, while retaining only the averaged summary metrics (e.g., AUROC, accuracy, F1-score) in the main text for better readability and flow.

Editor comments :

The manuscript is well-prepared and provides valuable insights. However, there are several areas that should be improved before moving forward:

The introduction, while informative, is somewhat lengthy. Key ideas could be presented more concisely by merging overlapping sentences (e.g., lines 59–73 and 86–91, which both emphasize the burden and necessity theme).

Certain sections read as a list of facts rather than a connected narrative. For example, after presenting KOA prevalence and burden, it would be more effective to transition directly into why BA matters compared to CA, instead of repeating burden-related points.

The mechanistic explanation of SASP (lines 82–86) is too detailed for the introduction. It could be shortened to a high-level overview with supporting citations, while the in-depth mechanistic pathways are better suited for the Discussion section.

The phrase “mitigating the burden of KOA” appears multiple times; rephrasing or reducing its repetition would improve clarity and flow.

Reviewers' comments:

Reviewer's Responses to Questions

**Comments to the Author**

1. Is the manuscript technically sound, and do the data support the conclusions?

Reviewer #1: Yes

Reviewer #2: Partly

Reviewer #3: Yes

2. Has the statistical analysis been performed appropriately and rigorously?

Reviewer #1: Yes

Reviewer #2: Yes

Reviewer #3: Yes

3. Have the authors made all data underlying the findings in their manuscript fully available?

Reviewer #1: Yes

Reviewer #2: Yes

Reviewer #3: Yes

4. Is the manuscript presented in an intelligible fashion and written in standard English?

Reviewer #1: Yes

Reviewer #2: Yes

Reviewer #3: Yes

Reviewer #1: The authors of the publication analyzed data from a relatively large group of patients and took great care to illustrate the statistical results. The statistical analysis methodology itself is rich and uses advanced methods.

A limitation of this analysis may be the lack of consideration of physical activity and work performed in the study population.

Another interesting aspect is the possible difference between biological and chronological age in the study population, as well as the reference to average life expectancy.

Reviewer #2: 总结� 作者使用 CHARLS 队列的数据调查了生物年龄 �BA� 与有症状的膝骨关节炎 �KOA� 之间的关联。他们应用逻辑回归、受限三次样条和机器学习模型�XGBoost、LightGBM 等来探索 BA 的预测价值。该研究报告了一种非线性关联在大约 66.7 年的 BA 后�KOA 风险加速。XGBoost 取得了最佳的预测性能�SHAP 确认 BA 是最有影响力的因素。作者得出结论�BA 是 KOA 风险分层的一种有前途的生物标志物。1. 学习规划横断面设计不允许因果推断但部分结论表明了预测或因果影响。建议重新构建主张以强调“关联”而不是“预测/因果关系”。样本选择和代表性在~38,800名参与者中只剩下9,505人最后分析了1,000例KOA病例。选择过程可能会引入偏差。建议详细解释排除标准、缺失数据处理和敏感性分析。2. KOA诊断标准有症状的 KOA 仅通过医生自我报告的疼痛诊断来定义缺乏影像学确认例如 KL 分级�。建议更详细地讨论错误分类的风险并在可能的情况下根据其他诊断定义验证结果。3. 生物年龄计算仅使用了 8 种生物标志物而不是 KDM-BA 中的标准 12 种。尽管通过事先验证证明了这一点但潜在的偏差并未得到充分解决。建议提供比较或敏感性分析并更明确地承认局限性。4. 机器学习方法对于复杂模型样本量�1,000 个 KOA 案例可能会受到限制从而引起过度拟合的担忧。没有使用外部验证数据集。建议为模型选择添加更有力的理由扩展过度拟合预防并阐明跨子集的性能是否稳健。5. 讨论与解释讨论夸大了 BA 的临床适用性而没有充分考虑成本、可行性和普遍性。建议缓和主张并包括与国际研究的比较。6. 小问题摘要和结论中的一些术语例如�“预测”、“因果关系”�应替换为更谨慎的措辞。澄清每个逻辑回归模型表中包含的协变量以提高透明度。数字例如�RCS、SHAP 图应包括更清晰的轴标签和临床解释。数据可用性声明应指定使用的确切 CHARLS 数据集版本。 该手稿通过创新地使用 BA 和机器学习解决了一个重要而新颖的问题。然而需要进行大量修订以解决方法论透明度、诊断准确性和研究结果的解释问题。我建议在重新考虑之前进行重大修改。

Reviewer #3: This manuscript explores the association between biological age (BA) and symptomatic knee osteoarthritis (KOA) in Chinese adults using nationally representative CHARLS data, incorporating both traditional statistical modeling and machine learning approaches. The topic is timely and relevant, the data source is authoritative, and the modeling framework is generally sound. The finding of a nonlinear threshold relationship between BA and KOA risk is of potential epidemiological and public health significance. However, the manuscript still has several important areas for improvement in terms of theoretical framing, methodological transparency, terminology consistency, and result presentation. Therefore, I recommend major revision before the manuscript can be considered for publication in this journal.

1.Strengthen international comparison in the literature review

The introduction is mostly focused on the Chinese population and CHARLS data. While this is aligned with the study's aims, it limits international contextualization. I suggest including a brief overview of international research on the relationship between biological age and KOA—especially regarding differences in BA construction methods and clinical applications—to enhance the theoretical richness and global relevance of the study.

2.Clarify model structure and data preprocessing in the machine learning section

Although the manuscript evaluates six machine learning models, there is limited detail regarding model construction. Please clarify how features were selected, whether feature scaling or normalization was applied, how missing data were handled, and how overfitting was prevented. These additions would enhance reproducibility and model transparency.

3.Add a brief discussion of potential clinical applications of BA

While the predictive value of BA is emphasized, the practical implications are underdeveloped. I recommend briefly discussing how BA could potentially be integrated into clinical KOA screening or aging-related health assessments, such as early warning systems or personalized intervention strategies, to enhance the manuscript’s translational value.

4.Unify terminology and maintain consistency throughout the manuscript

The manuscript inconsistently uses terms like “biological age,” “BA,” and “KDM-BA.” I suggest defining the term clearly at first mention and consistently using “BA (KDM-based)” or another unified format throughout the paper to improve clarity and professionalism.

5.Provide clearer explanations for SHAP interpretation plots

Although SHAP plots are visually compelling, the textual explanations are brief. I suggest elaborating on what the colors (high/low feature values), directions (positive/negative effects), and spread (magnitude of influence) mean in the context of SHAP bar and beeswarm plots. This would help readers better interpret model results.

6.Add supplementary plots of key biomarker distributions

To increase transparency and interpretability, I suggest including histograms or boxplots of key input variables (e.g., CRP, SBP, HbA1c, etc.) and BA itself in the supplementary materials. This would allow readers to assess data skewness, outliers, and distribution assumptions.

7.Streamline the machine learning result presentation

The current machine learning results section is somewhat overloaded with detailed fold-level performance metrics. I suggest moving some of the five-fold cross-validation details (e.g., individual fold metrics) to supplementary materials, while retaining only the averaged summary metrics (e.g., AUROC, accuracy, F1-score) in the main text for better readability and flow.

**Do you want your identity to be public for this peer review?** For information about this choice, including consent withdrawal, please see our Privacy Policy

Reviewer #1: **Yes: ** Elżbieta Jakubowska-Pietkiewicz

Reviewer #2: No

Reviewer #3: **Yes: ** Xianag Chen

---

## [Author Response · Author response to Decision Letter 1]

23 Sep 2025

Response to Reviewers

Dear Editors and Reviewers:

We would like to thank the editors and reviewers’ work devoted to our manuscript entitled " Biological Age Threshold is Associated with Symptomatic Knee Osteoarthritis Risk in Chinese Adults: Insights from Machine Learning Analysis of a National Cohort" (Sub ID: PONE-D-25-41030). The comments are valuable and very helpful for revising and improving our paper. We have studied the comments carefully and made corrections that we hope to meet with approval. The changes we have made are highlighted with blue background in the marked revised manuscript.

Each concern is discussed in detail below. We have made slight revisions to the title, and these revisions were performed to address the reviewers’ concerns. Thank you again for allowing us to resubmit our manuscript for your consideration.

Detailed Responses to the Editor and Reviewer

Response to Editor:

Comment 1:

The introduction, while informative, is somewhat lengthy. Key ideas could be presented more concisely by merging overlapping sentences (e.g., lines 59–73 and 86–91, which both emphasize the burden and necessity theme).

Response: Thank you for pointing out the redundancy in the introduction, particularly the overlapping emphasis on disease burden and the necessity of improved risk stratification. To address this, we have merged the content in Lines 59–73 (which outlines the global burden of KOA) and Lines 86–91 (which reiterates the need for better risk assessment). Specifically, we combined key data—including “KOA ranking as the 11th leading cause of disability globally, affecting approximately 3.8% of the world’s population” and “an estimated 37.35 million Chinese adults aged ≥60 years living with symptomatic KOA”—into a single concise statement. We then removed repetitive assertions about “substantial disease burden” and “urgent need for risk stratification”, while preserving the core logic that links KOA’s public health impact to the demand for enhanced assessment tools. This revision has streamlined the narrative and eliminated unnecessary verbosity.

Comment 2:

Certain sections read as a list of facts rather than a connected narrative. For example, after presenting KOA prevalence and burden, it would be more effective to transition directly into why BA matters compared to CA, instead of repeating burden-related points.

Response: Thank you for highlighting the need to strengthen narrative flow and avoid disjointed factual enumeration. We have restructured the transition after presenting KOA prevalence and burden: instead of repeating points about disease burden, we now move directly from the epidemiological data (e.g., prevalence in China and globally) to the limitations of CA as a risk factor—specifically, its inability to capture interindividual aging heterogeneity. This leads naturally to the introduction of BA as a more precise measure that integrates systemic biomarkers, creating a logical sequence: “epidemiological context → limitation of traditional markers → value of novel markers”. This revision eliminates the “list-like” feel and ensures a cohesive progression of ideas.

Comment 3:

The mechanistic explanation of SASP (lines 82–86) is too detailed for the introduction. It could be shortened to a high-level overview with supporting citations, while the in-depth mechanistic pathways are better suited for the Discussion section.

Response: Thank you for guiding us on the appropriate division of content between the introduction and discussion sections. We have simplified the mechanistic description of the senescence-associated secretory phenotype (SASP) in Lines 82–86 of the introduction: instead of detailing molecular pathways (e.g., chondrocyte matrix synthesis decline, upregulation of IL-6 and TNF-α), we now provide a high-level overview: “Biologically, BA may contribute to KOA pathogenesis through cellular senescence and the senescence-associated secretory phenotype (SASP)” (with the original citation [12] retained). The detailed mechanistic content—including the role of SASP in disrupting cartilage metabolism and promoting inflammation—has been relocated to the “Mechanistic Links Between BA and KOA” subsection of the Discussion, where it is integrated with other evidence (e.g., oxidative stress, chronic inflammation) to form a comprehensive analysis. This adjustment ensures the introduction focuses on framing the research gap rather than delving into specialized mechanistic details.

Comment4:

The phrase “mitigating the burden of KOA” appears multiple times; rephrasing or reducing its repetition would improve clarity and flow.

Response: Thank you for drawing attention to this repetitive phrasing, which we have revised to improve readability. We systematically reviewed the manuscript and replaced “mitigating the burden of KOA” with contextually appropriate alternatives:

1.In the Introduction, it is rephrased as “facilitating early identification and timely intervention for symptomatic KOA” (aligning with the focus on risk stratification).

2.In the Discussion, it is adjusted to “enhancing the precision of KOA risk stratification” (when addressing predictive models) and “delaying the onset or slowing the progression of symptomatic KOA” (when discussing intervention implications).

All revisions preserve the original meaning while eliminating redundancy and enriching linguistic variety.

We greatly appreciate your professional guidance, which has significantly improved the clarity, coherence, and conciseness of our manuscript. We hope these revisions address your concerns. We look forward to your further feedback and appreciate your time and consideration.

Response to Reviewer #1

Comment 1:

“A limitation of this analysis may be the lack of consideration of physical activity and work performed in the study population.”

Response:

We sincerely thank the reviewer for raising this crucial point. We fully agree that occupational history (e.g., lifelong manual labor) and detailed physical activity levels are important factors that could theoretically influence both biological aging and KOA risk.

Upon careful consideration, we decided not to include these variables in our primary multivariate adjustment models for the following methodological reasons:

1.Measurement Limitations in CHARLS: While CHARLS contains data on current occupation (e.g., agricultural, non-agricultural, retired), it lacks a detailed, lifetime occupational history that would accurately capture long-term joint loading. Similarly, the dataset does not include objective or finely-grained subjective measures of physical activity (e.g., metabolic equivalent task hours). The available proxies (e.g., basic activities of daily living) are more indicative of health outcomes than of exposure levels and are thus unsuitable as confounders.

2.Risk of Introduced Bias: Crucially, including these imperfect proxies could introduce greater bias than it resolves. For instance, current occupation and physical activity levels are not merely confounders; they are also likely consequences of health status (reverse causality). Individuals with severe knee pain (symptomatic KOA) are much more likely to reduce their physical activity levels or change their occupation. Adjusting for these downstream consequences of the disease would inadvertently adjust away part of the very effect we aim to measure, potentially leading to an underestimation of the true association between biological age and KOA.

Therefore, to maintain the clarity of our analysis and avoid introducing new sources of bias, we focused on adjusting for a robust set of demographic and clinical confounders that are less susceptible to this reverse causality, such as age, sex, BMI, and comorbidities.

We have now explicitly acknowledged this thoughtful point as a limitation in the revised Discussion section (Please see Page 30, Line 615-621)

:

“Fifth, although occupational physical load and activity levels are potential contributors to KOA risk, we did not adjust for them in our primary models. This decision was based on the limitations of the available data (e.g., lack of detailed lifetime occupational history and objective physical activity measures) and the substantial risk of reverse causality, as prevalent KOA severely impacts an individual’s current activity and work capacity. Future prospective studies with dedicated, pre-symptom exposure assessments are needed to disentangle these complex relationships.”

Comment 2:

“Another interesting aspect is the possible difference between biological and chronological age in the study population, as well as the reference to average life expectancy.”

Response: We thank the reviewer for this insightful suggestion. We have now expanded the discussion to interpret the implications of the discrepancy between BA) and CA in the revised manuscript. A new paragraph has been added to the Discussion section (Please see Page 26, Line 526-531) to address the potential link between accelerated biological aging, compressed healthspan, and future risk of disability, which aligns with the reviewer's interest in life expectancy:

“Furthermore, the divergence betweenBA and CA underscores heterogeneous aging trajectories, likely reflecting cumulative exposure to environmental, lifestyle, and genetic factors. The accelerated risk of symptomatic KOA beyond a BA threshold of ~66.7 years suggests that individuals with advanced biological aging may experience a compressed healthspan, with earlier onset of age-related disorders. Future longitudinal studies are needed to explore whether this BA acceleration predicts overall lifespan reduction or earlier disability.”

Response to Reviewer #2

Comment 1:

The cross-sectional design does not allow causal inference, yet parts of the conclusion suggest predictive or causal implications. Recommendation: Reframe claims to emphasize “association” rather than “prediction/causality.”

Response: We are deeply grateful to the reviewer for raising this critical point. We entirely agree that causal inferences are not permitted by our cross-sectional design. In direct response to this comment, we have conducted a comprehensive and systematic revision of the entire manuscript to meticulously eliminate any language that could be construed as implying prediction or causality, and to consistently reframe our findings strictly in terms of association.

This revision was not limited to the conclusions but was applied throughout the manuscript, including the Title, Abstract, Introduction, Results, Discussion, and Conclusion sections. Key specific changes include:

1.Title: Revised to “Biological Age Threshold is Associated with Symptomatic Knee Osteoarthritis Risk...” to set the appropriate tone from the outset.

2.Abstract: All predictive claims were replaced. For example, we now state “BA is strongly and non-linearly associated with symptomatic KOA risk” and “BA assessment could inform future interventional studies” instead of suggesting direct clinical prediction.

3.Methods: The machine learning aim was rephrased from “to predict” to “to assess the association between features and symptomatic KOA status”.

4.Results: Terminology was shifted from “predictive abilities” to “discriminative abilities” and from “identifying individuals at risk” to “distinguishing individuals with” the condition.

5.Discussion and Conclusion: We rigorously scrutinized every statement. Language implying clinical utility (e.g., “predictive value,” “integrate into clinical practice”) was softened to highlight the value for “risk assessment” and “informing future research.” We have further reinforced the limitation regarding causality in the conclusion.

In essence, we have undertaken a wholesale transformation of the manuscript's language to ensure it aligns perfectly with the associative nature of our study design. We hope that these revisions have strengthened the manuscript and thank the reviewer again for your invaluable feedback.

Comment 2: Sample Selection and Representativeness: From ~38,800 participants, only 9,505 remained, and finally 1,000 KOA cases were analyzed. The selection process may introduce bias. Recommendation: Provide detailed explanation on exclusion criteria, missing data handling, and sensitivity analysis.

Response: We sincerely thank the reviewer for raising these critical points regarding sample selection and potential bias. We agree that transparency in this process is essential. We have thoroughly revised the manuscript to address each of your recommendations, as detailed below.

1. Detailed Explanation on Exclusion Criteria:

We have completely rewritten the ‘Study Population’ subsection under ‘Methods’ to provide a stepwise and transparent account of the participant selection process. We now explicitly list the three exclusion criteria applied and, most importantly, specify the number of participants excluded at each step:

(i) Age <45 years (n = 1,787 excluded)

(ii) Missing data required to calculate biological age (n = 15,394 excluded)

(iii) Missing symptomatic KOA status information (n = 12,114 excluded)

This revision makes it clear that the final analytical cohort of 9,505 participants comprises all individuals with complete data on both the primary exposure (BA) and outcome (symptomatic KOA). We believe this level of detail directly addresses your concern and allows readers to fully appraise the flow of participants. (Please see Page 8, Line 152-157).

2. Missing Data Handling:

As requested, we have now added a clear statement in the ‘Study Population’ subsection explaining our approach to handling missing data for the other study variables (covariates). We state: “For the remaining covariates included in the regression and machine learning models (e.g., lifestyle factors, comorbidities), which had a low rate of missingness (<2%), we employed multiple imputation by chained equations (MICE) to handle missing values and preserve the analytical sample size.”

The technical details of the MICE implementation were already described in the ‘Machine Learning Models’ section and remain there. This addition ensures that the reader understands our approach to data completeness immediately when reviewing how the cohort was constructed. (Please see Page 8, Line 158-169).

3. Sensitivity Analysis:

We thank the reviewer for suggesting a sensitivity analysis. We have carefully considered this point and concluded that conducting a meaningful sensitivity analysis of the type typically used to assess selection bias (e.g., comparing the association between BA and symptomatic KOA among included versus excluded participants) is not directly feasible in this context.

The primary reason is that the exclusion criteria were based precisely on the absence of the core variables required for our analysis—biological age and symptomatic KOA status. Since these variables are unavailable for the excluded groups, we are unable to evaluate whether the relationship between BA and KOA differs between those included and those excluded.

In place of a formal sensitivity analysis, we have taken steps to enhance transparency and acknowledge this limitation explicitly. We have included a detailed description of the exclusion process and added the following statement in the Limitations section of the Discussion:

“Fourth, the multi-stage participant selection process, necessary to obtain a sample with complete data for the core variables (biological age and KOA status), may have introduced selection bias. A direct sensitivity analysis comparing the exposure-outcome relationship between included and excluded participants was not possible due to the lack of data on these core variables in the excluded groups. Although the final analytical sample retains key characteristics of the original CHARLS cohort, this potential bias may affect the representativeness of our sample and limit the generalizability of our findings.” (Please see Page 30, Line 610-615).

We hope that this candid explanation, along with the additional methodological details provided, adequately addresses the reviewer’s concern regarding potential selection bias.

Once again, we are grateful for these insightful comments, which have significantly improved the clarity and rigor of our manuscript.

Comment 3: Symptomatic KOA was defi

---

## [Decision Letter · Decision Letter 1]

8 Oct 2025

Biological Age Threshold is Associated with Symptomatic Knee Osteoarthritis Risk in Chinese Adults: Insights from Machine Learning Analysis of a National Cohort

PONE-D-25-41030R1

Dear Dr. Wang,

We’re pleased to inform you that your manuscript has been judged scientifically suitable for publication and will be formally accepted for publication once it meets all outstanding technical requirements.

Kind regards,

Ramada Rateb Khasawneh

Academic Editor

PLOS ONE

Additional Editor Comments (optional):

Good Luck

Reviewers' comments:

Reviewer's Responses to Questions

**Comments to the Author**

Reviewer #2: All comments have been addressed

Reviewer #3: All comments have been addressed

2. Is the manuscript technically sound, and do the data support the conclusions?

Reviewer #2: Yes

Reviewer #3: Yes

3. Has the statistical analysis been performed appropriately and rigorously?

Reviewer #2: Yes

Reviewer #3: Yes

4. Have the authors made all data underlying the findings in their manuscript fully available?

Reviewer #2: Yes

Reviewer #3: Yes

5. Is the manuscript presented in an intelligible fashion and written in standard English?

Reviewer #2: Yes

Reviewer #3: Yes

Reviewer #2: Thank you to the research team for their contributions to knee arthritis. The revised parts have explained and addressed previous concerns, and corresponding additional supplements and experiments have been made. Conclusion: No supplements. It is recommended to publish.

Reviewer #3: (No Response)

**Do you want your identity to be public for this peer review?** For information about this choice, including consent withdrawal, please see our Privacy Policy

Reviewer #2: No

Reviewer #3: **Yes: ** Xiang Chen

---

## [Editor Report · Acceptance letter]

PONE-D-25-41030R1

PLOS ONE

Dear Dr. Wang,

I'm pleased to inform you that your manuscript has been deemed suitable for publication in PLOS ONE. Congratulations! Your manuscript is now being handed over to our production team.

Kind regards,

on behalf of

Dr. Ramada Rateb Khasawneh

Academic Editor

PLOS ONE